# PeerJ

# Factors associated with leisure time physical inactivity in black individuals: hierarchical model

Francisco José Gondim Pitanga[1], Ines Lessa[2], Paulo José B. Barbosa[3], Simone Janete O. Barbosa[4], Maria Cecília Costa[5] and Adair da Silva Lopes[6]

[1] Department of Physical Education of the Faculty of Education of Universidade Federal da Bahia (UFBA), Salvador, Bahia, Brazil
[2] Collective Health Institute of Universidade Federal da Bahia (UFBA), Brazil
[3] Universidade do Estado da Bahia, Brazil
[4] União Metropolitana de Educação e Cultura, Brazil
[5] Escola de Nutrição da Universidade Federal da Bahia (UFBA), Brazil
[6] Universidade Federal de Santa Catarina (UFSC), Brazil

## ABSTRACT

**Background.** A number of studies have shown that the black population exhibits higher levels of leisure-time physical inactivity (LTPI), but few have investigated the factors associated with this behavior.

**Objective.** The aim of this study was to analyze associated factors and the explanatory model proposed for LTPI in black adults.

**Methods.** The design was cross-sectional with a sample of 2,305 adults from 20–96 years of age, 902 (39.1%) men, living in the city of Salvador, Brazil. LTPI was analyzed using the International Physical Activity Questionnaire (IPAQ). A hierarchical model was built with the possible factors associated with LTPI, distributed in distal (age and sex), intermediate 1 (socioeconomic status, educational level and marital status), intermediate 2 (perception of safety/violence in the neighborhood, racial discrimination in private settings and physical activity at work) and proximal blocks (smoking and participation in Carnival block rehearsals). We estimated crude and adjusted odds ratio (OR) using logistic regression.

**Results.** The variables inversely associated with LTPI were male gender, socioeconomic status and secondary/university education, although the proposed model explains only 4.2% of LTPI.

**Conclusions.** We conclude that male gender, higher education and socioeconomic status can reduce LTPI in black adults.

Corresponding author
Francisco José Gondim Pitanga,
pitanga@lognet.com.br

## INTRODUCTION

Leisure-time physical inactivity (LTPI), defined as non-participation in activities involving body movements during free time, is associated with different metabolic and cardiovascular disorders in adults from different ethnic groups (*Kurian & Cardarelli, 2007*; *Pitanga & Lessa, 2009*). A number of studies have demonstrated that the black

population exhibits higher levels of LTPI, but few have investigated factors associated with this behavior (*Marshall et al., 2007*; *Ahmed et al., 2005*).

In Brazil the population is predominantly mixed race, with 49% black (mulattos + black), which has never been studied separately for LTPI. Salvador, the third largest city in Brazil, with a 70% black population (blacks + mulattos), is the most propitious urban environment for investigating this ethnicity (Brazilian Institute of Geography and Statistics (IBGE); *Lessa et al., 2006*). Moreover, in Brazil, these people have historically been discriminated against by society, which can cause important inequalities in different health variables, including physical activity behavior (*Kurian & Cardarelli, 2007*).

Even though socioeconomic level, schooling and age are reported in the literature as possible determinants of LTPI (*Marshall et al., 2007*; *Ahmed et al., 2005*; *Marquez, Neighbors & Bustamante, 2010*; *Pitanga & Lessa, 2005*), only one Brazilian study analyzed these variables in black adults. However, this study used leisure-time physical activity (LTPA) as outcome, showing a positive association with male gender, as well as higher schooling and socioeconomic levels (*Pitanga et al., 2012*).

Furthermore, variables such as racial discrimination and perception of violence, or fear in the neighborhood are considered potential determinants of ethnic-racial disparities existing in health and may be associated with LTPI (*Shelton et al., 2009*; *Roman et al., 2009*; *Piro, Noss & Claussen, 2006*). Another possible determinant of LTPI is physical activity at work (PAW), since individuals with a physically active work day may not be inclined to engage in physical activity in their free time. However, there is still no evidence to confirm these speculations, primarily in black adults (*Marquez, Neighbors & Bustamante, 2010*).

On the other hand, smoking may also be associated with LTPI, considering the evidence that adults who stopped smoking after attending anti-smoking clinics significantly increased their physical activity (*Hassandra et al., 2012*).

Finally, given that the Carnival is an integral part of the culture of Salvador, Bahia, it is also necessary to determine if taking part in Carnival block rehearsals contributes to reducing LTPI, since these rehearsals occur throughout the year and are widely attended by black adults.

Different sophisticated techniques have been used in an attempt at elucidating these questions. The hierarchical model has been proposed to analyze different factors that determine health conditions or disease (*Victora et al., 1997*). A number of Brazilian articles have used this model to explain physical inactivity in population groups, but not specifically in the black population (*Florindo et al., 2009*; *Fonseca et al., 2008*).

Thus, it is important to identify the main determinants, and propose an explanatory theoretical model for LTPI in black adults. This information can be used to make public health managers aware of the importance of encouraging physical activity, since it can be used as one of the means of preventing metabolic and cardiovascular disorders, thereby decreasing excessive spending on the most complex services provided by the health system (*Pitanga & Lessa, 2008*).

Thus, the aim of the present study was to analyze associated factors and propose an explanatory model for LTPI in black adults.

## METHODOLOGY

### Design and study site

This is a cross-sectional study conducted in Salvador, Brazil in 2007, with a focus on non-transmissible chronic diseases and their risk factors. The city of Salvador is subdivided into 12 sanitary districts (SD), four of which have a black population of 75%. Two of these SD were selected by convenience, both densely populated: Liberdade, with its seven neighborhoods, and Barra-Rio Vermelho with 56% of its neighborhoods.

### Sampling

The basis for the sample size was the 35% prevalence of hypertension observed in black subjects in a study conducted in Salvador in 2006 that included all ethnicities (*Victora et al., 1997*). Considering an error of less than 2% and 95% confidence level, the estimate was 2,185 ($\approx$2,200) black adults aged $\geq$20 years. As a general rule subjects were interviewed at home. Thus, the number of households randomly selected was based on the number of participants, but on rare occasions when more than one non-blood related family lived in the same house, one eligible subject per family was drawn.

Since blacks account for 75% of the population in the selected areas, the same should be true for the general population. Considering a 25% white population, it would be necessary to visit 2,950 households, 2,200 of these inhabited by blacks and $\approx$750 by whites, with the latter disregarded for study purposes. A total of 740 households ($\approx$25%) were added to cover unoccupied and non-residential dwellings, as well as those inhabited by individuals younger than 20 years of age or absent at two consecutive visits, increasing the sample size to 3,690. A further 15% (553) were added to compensate for household or individual refusals. Thus, the total number of households was estimated at 4,243, rounded off to 4,250, probabilistically drawn from all streets.

First, a census of the entire area was used to delimit the streets and count the residences. Next, with the help of maps, simple random samples were extracted from (a) streets; (b) residences on the streets drawn ($n = 4,250$); and (c) an eligible individual from households inhabited by one family or two individuals from more than one non-blood related family. The number of households sampled foresaw the exclusion of 25% of white residences and 25% for unoccupied dwellings, absent inhabitants, ineligible individuals, household and individual refusals, non-residential or abandoned buildings and vacant lots.

### Elegibility

The following eligibility criteria were adopted: subjects who refer to themselves as black or mulatto, age $\geq$20 years and willing to take part in both stages of the investigation: (1) household survey and (2) appearing at the health facility for complementary examinations. Individuals who declared themselves white, pregnant women or those without the mental capacity to respond to the questionnaire or to appear at the health facility for the second stage of the study were ineligible. Before the participant draw, residents were questioned as to skin color, using only the options adopted in censuses

undertaken by the IBGE, and if they accepted to participate in both phases of the study. Those who agreed to undergo examinations were admitted to the sampling process. If the sampled individual refused to submit to the complementary examinations, it was considered a refusal and the person was excluded from the investigation.

## Data collection instrument

All the study participants were interviewed at home for collection of sociodemographic and physical activity data. The data collection instrument was a questionnaire programmed in Java, for use on a Palm Z22 PDA. To that end, the instrument was planned and discussed with the project team and then with the computer programmer, in order to fit it to the program and discuss the different types of responses and coding. When the questions were interdependent, most had an information control key (accept/reject), precluding any delay in entering information. Furthermore, no new questionnaires could be scheduled if the next to last of them had not been completed and saved. Weekly, or even beforehand if necessary, one of the project coordinators received the devices and transferred the questionnaire data to a computer, directly to the Excel program. The device can store up to 100 questionnaires (163 questions with innumerable possibilities) and the Palm of each interviewer could hold 100 questionnaires with consecutive numbers. After 100 were completed, another 100 were installed. Training in the use of the Palm was conducted by the programmer, initially for the team of coordinators and later for the ten interviewers in the presence of the entire team. After training, the pilot test was applied. Each interviewer, all with secondary schooling and extensive interviewing experience, applied ten questionnaires. The pilot study also functioned to test the performance of the Palm, its ease of use and duration of the interview, which was automatically recorded by the program. The interviewers were supervised in the field by higher education technicians.

## Study variables

The following variables were used:
 Dependent variable: LTPI
 Independent variables in hieracrchy:

1. Demographic variables (distal)
   - Sex
   - Age
2. Social variables (intermediate)
   - Socioeconomic level
   - Schooling
   - Marital status
   - Perception of safety in the neighborhood
   - Racial discrimination in private settings
   - Physical activity at work

3. Behavioral variables (proximal)
   - Smoking
   - Participation in carnival block rehearsals

To identify physical activity the long form of the International Physical Activity Questionnaire (IPAQ) was used. This instrument is composed of questions regarding the frequency and duration of physical activities (walking, moderate and vigorous) performed during the last week and engaged in at work, commuting, domestic activities and leisure-time (*Matsudo et al., 2001*). Physical activity values were reported in minutes/week by multiplying the weekly frequency by the duration of each activity performed. This study used only the physical activity during leisure-time and at work domains. LTPI was categorized as 0 = physically active (≥150 min per week on moderate physical activities or walking and/or ≥60 min per week of vigorous physical activities) and 1 = physically inactive (<150 min per week on moderate physical activities or walking and/or <60 min per week on vigorous physical activities). Physical activity at work (PAW) was characterized as 0 = physically inactive (<150 min per week on moderate physical activities or walking and/or <60 min per week on vigorous physical activities) and 1 = physically active (≥150 min per week on moderate physical activities or walking and/or <60 min per week on vigorous physical activities).

Three strata were established for schooling: 0 = very low (illiterates to fifth graders); 1 = low (elementary schooling); 2 = middle/high (secondary schooling, including professional technical courses and complete or incomplete university education). Social classes were classified, according to the Brazilian Association of Market Research (ABPEME-IBGE), into A to E. In this study they were classified as follows: low = 0 (classes D + E); middle/high = 1 (classes A + B + C). For marital status the following classification was adopted: 0 = single, 1 = married or in a common law relationship, 2 = separated, divorced or widowed. The following classification was adopted for age: age = 0 if <40 years, age = 1 between 40 and 59 years and age = 2 if ≥60 years. Racial discrimination in private settings was self-reported and classified as follows: 0 = no and 1 = yes. Perception of policing in the neighborhood was also self-reported and classified according to the following criteria: 0 = yes and 1 = no. Perception of violence in the neighborhood, also self-reported, adopted the following classification: 0 = very peaceful, very good place to live, 1 = not very peaceful, but a good place to live, 2 = bad place to live, with threats (of any kind) and not peaceful: it is violent with street fights and/or armed people, drug users or drug dealers. Smoking was classified as follows: 0 = non-smoker; 1 = smoker or ex-smoker, and participation in Carnival block rehearsals was classified as: 0 = non-participant; 1 = participant.

## Analysis procedures

Variable characterization was presented as prevalences and their respective 95% confidence intervals. Next, OR were estimated by conducting univariate and multivariate analyses, using logistical regression, based on a previously defined theoretical model that discriminates the potential associated factors into hierarchized blocks (Fig. 1), in line

**Distal Block**

DEMOGRAPHIC
VARIABLES
Age, Sex

**Intermediate block 1**

SOCIAL VARIABLES 1
Socioeconomic level,
Schooling, Marital Status

**Intermediate Block 2**

SOCIAL VARIABLES 2
Perception of
safety/violence in the
neighborhood, Racial
discrimination in private
settings, Physical activity
at work

**Proximal Block**

BEHAVIORAL VARIABLES
Smoking, Participation in Carnival
block rehearsals

LEISURE-TIME
PHYSICAL
INACTIVITY

**Figure 1  Multivariate hierarchical model for analysis of factors associated to leisure-time physical inactivity in black adults.**

with the hierarchy existing between the levels of determination of LTPI. The strategy used for inputting blocks of variables was the step forward method, as follows: distal block (demographics); intermediate block 1 (socioeconomic 1); intermediate block 2 (socioeconomic 2); proximal block (behavioral). Variables showing statistically significant levels, according to a *p*-value <0.20 remained in the model. A 95% confidence interval (CI) level was adopted as well as STATA 7.0 statistical software.

The Project was approved by the Ethics Committee of Instituto de Saúde Coletiva da UFBA, protocol no. 002-07 and the study was conducted in accordance with standards required by the Declaration of Helsinki, with no conflicts of interest in its content. All study participants gave their informed consent.

## RESULTS

Out of the total sample, 1.2% of eligible individuals refused to undergo complementary examinations (partial refusal) and were excluded, but two-thirds of these refusals were spontaneously reverted during the investigation. However, there was a 4.6% surplus of participants over the expected number, resulting in a final sample of 2,305 blacks, 902 (39.3%) men, who agreed to take part in both stages of the research.

The statistical power of this sample to identify the associations between study variables and LTPI was calculated later, considering LTPI prevalence among those not exposed of 15%, confidence interval of 95%, power of 80% and odds ratio (OR) less than or equal to 0.53.

The prevalences of physical inactivity and the different variables analyzed in the present study are demonstrated in Table 1. There was a larger proportion of women, subjects aged between 20 and 59 years, low socioeconomic status, very low or medium/high schooling and married individuals. With respect to racial discrimination, most of the participants reported not feeling discriminated against in private settings. In regard to policing/violence in the neighborhood, most of the individuals considered the neighborhood as not very peaceful and poorly policed. In relation to physical inactivity, there was higher prevalence at work than in leisure time.

Table 2 illustrates crude and adjusted OR between LTPI and the variables of the different blocks in hierarchical analysis. Inverse associations were found for male gender, socioeconomic status and medium/high schooling levels.

Table 3 demonstrates the contribution of each block of variables associated to LTPI for model fit. The entire model explains only 4.2% of LTPI.

## DISCUSSION

This study proposed to identify the factors associated with LTFI in black adults using a hierarchical model. The variables analyzed in the distal block (sex and age) and intermediate block 1 (socioeconomic level, schooling and marital status) showed an inverse association with the male gender, socioeconomic level and medium/high schooling.

In Brazil no studies were found on the factors associated to LTPI in black adults (*Pitanga & Lessa, 2005*), but one paper was found on PAW, demonstrating a positive association with male gender, higher schooling levels and higher socioeconomic status. Another

**Table 1** Prevalence of study variables. Salvador–Bahia–Brazil, 2014.

| Variables | n | % (CI 95%) |
|---|---|---|
| **Sex** | | |
| Female | 1,403 | 60.9 [52.2-63.4] |
| Male | 902 | 39.3 [35.9–42.4] |
| **Age** | | |
| 20–39 | 973 | 42.2 [39.1–45.4] |
| 40–59 | 949 | 41.2 [38.0–44.4] |
| ≥60 | 383 | 16.6 [13.1–20.8] |
| **Socioeconomic Level** | | |
| Low | 1,560 | 67.7 [65.3–70.0] |
| Medium/High | 745 | 32.3 [29.0–35.8] |
| **Schooling** | | |
| Very low | 913 | 39.6 [36.5–42.9] |
| Low | 422 | 18.3 [14.7–22.2] |
| Medium/High | 970 | 42.1 [38.9–45.2] |
| **Marital status** | | |
| Single | 725 | 31.4 [28.1–35.0] |
| Married | 1,134 | 49.2 [46.3–52.20 |
| Separated/Widowed | 446 | 19.3 [15.7–23.3] |
| **RDPS** | | |
| Yes | 238 | 10.3 [6.9–15.1] |
| No | 2,067 | 89.7 [88.2–90.9] |
| **Perception of violence in the neighborhood** | | |
| Very peaceful | 568 | 24.6 [21.1–28.4] |
| Somewhat peaceful | 1,390 | 60.3 [57.7–62.9] |
| Violent/Very violent | 347 | 15.1 [11.4–19.2] |
| **Perception of the existence of policing in the neighborhood** | | |
| Yes | 606 | 26.3 [22.8–29.9] |
| No | 1,699 | 73.7 [71.5–75.8] |
| **Participation in Carnival block rehearsals** | | |
| Yes | 234 | 10.1 [6.7–14.9] |
| No | 2,071 | 89.9 [88.5–91.1] |
| **Smoking** | | |
| Non-smoker | 1,808 | 78.4 [76.4–80.3] |
| Smoker/Ex smoker | 497 | 21.6 [18.0–25.4] |
| **Physical Activity at Work** | | |
| Active | 221 | 9.6 [5.9–14.2] |
| Inactive | 2,084 | 90.4 [89.1–91.6] |
| **Leisure-Time Physical Inactivity** | | |
| Active | 257 | 11.2 [7.7–15.8] |
| Inactive | 2,048 | 88.8 [87.4–90.2] |

**Notes.**

$CI_{95\%}$, 95% confidence interval; RDPS, racial discrimination in private settings.

**Table 2** Association between LTPI and variables of different blocks of the hierarchical analysis of black adults. Salvador, Brazil, 2014.

| Variables | Crude odds ratio (95% CI) | Adjusted odds ratio (CI 95%) | p-value |
|---|---|---|---|
| **Block 1 (Distal)** | | | |
| **Sex**[a] | | | |
| Female | 1 | 1 | |
| Male | 0.41 [0.31–0.54] | 0.40 [0.31–0.52] | 0.00 |
| **Age**[a] | | | |
| 20–39 | 1 | 1 | |
| 40–59 | 0.89 [0.66–1.20] | 0.82 [0.61–1.10] | 0.19 |
| ≥60 | 0.82 [0.57–1.22] | 0.72 [0.49–1.05] | 0.08 |
| **Block 2 (Intermediate 1)** | | | |
| **Socioeconomic Level**[b] | | | |
| Low | 1 | 1 | |
| Medium/High | 0.66 [0.50–0.87] | 0.74 [0.56–0.99] | 0.05 |
| **Schooling**[b] | | | |
| Very low | 1 | 1 | |
| Low | 0.94 [0.63–1.42] | 0.87 [0.57–1.31] | 0.50 |
| Medium/high | 0.71 [0.53–0.96] | 0.65 [0.46–0.92] | 0.01 |
| **Marital status**[b] | | | |
| Single | 1 | 1 | |
| Married | 0.96 [0.70–1.32] | 1.02 [0.75–1.39] | 0.90 |
| Separated | 0.89 [0.61–1.32] | 0.75 [0.49–1.14] | 0.18 |
| **Block 3 (Intermediate 2)** | | | |
| **Perception of violence in the neighborhood**[c] | | | |
| Very peaceful | 1 | | |
| Somewhat peaceful | 1.28 [0.94–1.75] | 1.22 [0.90–1.67] | 0.21 |
| Violent/Very violent | 1.10 [0.72–1.70] | 1.07 [0.70–1.63] | 0.78 |
| **Perception of the existence of policing in the neighborhood**[c] | | | |
| Yes | 1 | 1 | |
| No | 1.15 [0.85–1.55] | 1.09 [0.81–1.47] | 0.57 |
| **Racial discrimination in private settings**[c] | | | |
| No | 1 | 1 | |
| Yes | 0.60 [0.41–0.90] | 0.90 [0.57–1.37] | 0.63 |
| **Physical activity at work**[c] | | | |
| No | 1 | | |
| Yes | 1.03 [0.66–1.69] | 1.24 [0.78–1.96] | 0.36 |
| **Block 4 (Distal)** | | | |
| **Smoking**[d] | | | |
| Non-smoker | 1 | 1 | |
| Smoker/Ex smoker | 0.93 [0.68–1.30] | 1.05 [0.76–1.46] | 0.76 |

Table 2 (*continued*)

| Variables | Crude odds ratio (95% CI) | Adjusted odds ratio (CI 95%) | *p*-value |
|---|---|---|---|
| **Participation in Carnival block rehearsals**[d] | | | |
| No | 1 | 1 | |
| Yes | 0.74 [0.49–1.13] | 0.86 [0.57–1.30] | 0.48 |

**Notes.**
[a] Adjusted for distal block variables.
[b] Adjusted for distal block and intermediate 1 variables.
[c] Adjusted for distal block, intermediate 1 and intermediate 2 variables.
[d] Adjusted for distal block and intermediate 1 variables.

**Table 3 Contribution of each block of variables associated to LTPI to fit the model.**

| Block of variables | Functional deviation | Chi-squared | *P*-value | Explicative power |
|---|---|---|---|---|
| Block 1 (sex + age) | −782.343 | 47.12 | 0.00 | 2.9% |
| Block 1 (sex + age) + Block 2 (NSE + Escolaridade + Estado Civil) | −773.430 | 64.95 | 0.00 | 4.0% |
| Block 1 (sexo + idade) + Block 2 (Socioeconomic status + Schooling + Marital Status) + Block 3 (Perception of safety/violence in the neighborhood, Racial discrimination in private settings, Physical Activity at work) | −768.559 | 74.69 | 0.00 | 4.2% |
| Block 1 (sex + age) + Block 2 (Socioeconomic status + Marital Status) + Block 4 (Smoking + Participation in Carnival block rehearsals) | −769.771 | 72.27 | 0.00 | 4.2% |

study conducted in Salvador (*Marquez, Neighbors & Bustamante, 2010*) analyzed the general population, where the prevalence of LTPI was lower than in this study of the black population. It was also observed that male gender and age were inversely associated to physical inactivity. These results were likely obtained because of the lower possibility of individuals with low socioeconomic and schooling levels engaging in leisure-time physical activities, in addition to the fact that men are more available to take part in these activities, mainly on weekends.

In the USA (*Marshall et al., 2007*) 4,695 adult men and 6,516 women were analyzed in order to identify the prevalence of LTPI in ethnic/racial groups between different indicators of social class. Social class indicators were schooling, family income, occupation and marital status. Corroborating the results of our study, the prevalence of LTPI within each ethnic/racial group was lower in the higher social classes.

In another study carried out in the USA (*He & Baker, 2005*), LTPI declined with higher schooling levels, a similar finding to that obtained here.

LTPI was also analyzed in a representative sample in the USA (*Ahmed et al., 2005*) composed of 23,459 male adults of different etnicities, where it was found that the likelihood of engaging in leisure-time physical activities is associated with being young, having higher education levels and income, owning your own home and having a better perception of health status. In the present study, schooling and socioeconomic level were associated with LTPI.

In relation to intermediate block 2 variables (perception of safety in the neighborhood, racial discrimination in private settings and physical activity at work), no associations were observed with any of these variables. Racial discrimination in Salvador, a city with a high percentage of black residents, is likely very rare.

Although discrimination is a potential determinant of ethnic/racial disparities in health, a small number of studies have been conducted to investigate whether it contributes to disparities in physical activity levels in population groups. In accordance with our results, a recent study performed in Boston with 1,055 predominantly black and hispanic individuals (*Shelton et al., 2009*) found no association between discrimination and physical activity.

The hypothesis that PAW would be associated to LTPI was also not confirmed in the present study. It is speculated that individuals with a physically active job may not be inclined to engage in leisure-time physical activities; however, no association was demonstrated between PAW and LTPI. In accordance with the results of this study and with the aim of examining the relationship between physical activity at work and during leisure time among ethnic/racial groups, data from 2000 to 2003 were gathered from the National Health Interview Survey (NHIS) (*Lessa et al., 2006*). It was found that LTPI was not associated to any ethnic/racial group.

Another variable that was not associated to LTPI was the perception of violence/security in the neighborhood. In this respect, it is important to underscore that few studies have investigated the association between perception of violence and physical activity. In a recent study conducted in the USA with a sample composed of 328 African Americans (*Roman et al., 2009*), aimed at analyzing the environmental indicators of fear of crime and their relationship with physical activity, perception of violence was associated with fear and physical activity, a finding not observed in the present study.

In regard to proximal block variables (smoking and participation in Carnival block rehearsals), no association was demonstrated with LTPI. Taking part in Carnival block rehearsals was not associated with LTPI, likely because participation is often not significant enough to classify individuals as active in their free time. With respect to smoking, the fact of being a smoker could also be associated to LTPI, considering that this could reduce the participation of individuals in physical activities.

A number of studies have analyzed the determinants of physical inactivity using the principles of social justice. One of these, *Lee & Cubbin (2009)* suggests that racial discrimination, the environment, positive attitudes towards health, among other variables, might explain physical inactivity in population groups.

A likely limitation of this study was the fact that only the black population was analyzed. This made it difficult to compare the results with individuals from the same population, given that only one Brazilian study that specifically investigated this population was found (*Pitanga et al., 2012*). Moreover, even though the questionnaire is a widely used instrument for analyzing physical activity in epidemiological research, its use may result in biased information, since it requires recording information directly from the subjects interviewed.

On the other hand, considering that the sampling of streets, households and participants was probabilistic, the minimum sample loss, standardized interviewers and procedures lead us to assume the internal validity of the study for the population with the eligibility characteristics described. However, caution should be taken when assuming the external validity of the study. This is because the sample was only extracted from city districts with the highest proportion of blacks, encompassing a large number of neighborhoods, and because the information could not be extrapolated to entire neighborhoods since it is known that 25 to 30% of the population is other than black.

Finally, using self-reports of perceived violence/security in the neighborhood may have biased the results. It is suggested that homicide rates in the neighborhoods be used in future studies as a variable representable of violence.

## CONCLUSIONS

The results of the present study suggest that male gender, socioeconomic status and medium/high schooling level are inversely associated with LTPI in black adults. Given that the final model explains only 4.2% of physical inactivity, additional research is recommended to analyze other demographic, social, environmental, behavioral and biological vairables as possible determinants of LTPI. Moreover, although we found an association between physical activity and violence suggest future studies that investigate the relationship between these variables.

### Funding

This project was financed by the National Council for Scientific and Technological Development (CNPq)/Ministry of Health, Brazil–Process no 09804/2006-1. The funders had no role in study design, data collection and analysis, decision to publish, or preparation of the manuscript.

### Grant Disclosures

The following grant information was disclosed by the authors:
National Council for Scientific and Technological Development (CNPq)/Ministry of Health, Brazil: 09804/2006-1.

### Competing Interests

There are no potential conflicts of interest among the authors of this work.

### Author Contributions

- Francisco José Gondim Pitanga performed the experiments, analyzed the data, contributed reagents/materials/analysis tools, wrote the paper, prepared figures and/or tables.
- Ines Lessa conceived and designed the experiments, performed the experiments, contributed reagents/materials/analysis tools, reviewed drafts of the paper.
- Paulo José B. Barbosa performed the experiments, reviewed drafts of the paper.

- Simone Janete O. Barbosa and Maria Cecília Costa performed the experiments, contributed reagents/materials/analysis tools, reviewed drafts of the paper.
- Adair da Silva Lopes analyzed the data, reviewed drafts of the paper.

## Human Ethics

The following information was supplied relating to ethical approvals (i.e., approving body and any reference numbers):

The Project was approved by the Ethics Committee of Instituto de Saúde Coletiva da UFBA, protocol no. 002-07 and the study was conducted in accordance with standards required by the Declaration of Helsinki, with no conflicts of interest in its content. All study participants gave their informed consent.

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
