# Peer review of "Factors associated with leisure time physical inactivity in black individuals: hierarchical model"

_PeerJ, doi:10.7717/peerj.577_

## Round 0.1 · original submission · Minor Revisions

· Academic Editor

Minor Revisions

The reviewers have rightly pointed out places in which you could improve the writing somewhat.

Most importantly, you have only explained a small amount of variance so you need to describe what you have overlooked to guide future work. Also please explain why you are analyzing data collected 7 years ago and what impact the old age of the data may have for future research.

·

Basic reporting

The study reported in this paper is quite impressive, in aims, scope and methodology. The writing is fine, except for a few minor errors and typos. The results are disappointing, since such a small part of the LTPI variance was explained by the independent variables measured. I would recommend extending the discussion to speculate on what may be more influential variables. Could it be that person variables (such as personality or subjective health) have greater explanatory power? Possibly the authors have other thoughts on this matter, and it would make the paper more interesting.The reader is left without a sense of where the research should head now.

Experimental design

No comments

Validity of the findings

No Comments

Additional comments

Just to polish the manuscript there are a few places in the paper where two words are merged together without a space; in one place "anos" appears instead of "years".

·

Basic reporting

The title of the article could potentially be more descriptive of the study by adding that the individuals live in Brazil or in a single city in Brazil.

There are some statements that are difficult to understand on multiple readings. I have made annotations at these areas. Figures are overall well done.

Experimental design

The method of selection and efforts taken to collect a representative sample were adequate. It seems that the IPAQ was used to account for general physical activity level although it only appears to question activities over the previous 7 days which could lead to some error in reporting overall physical activity. Methods were descriptive.

Validity of the findings

The conclusions are well supported by the data. The results seem to conform with prior studies. It is not entirely clear what new information this particular study might add to the existing literature. The investigating the relationship between perceived violence and physical activity appears novel and is something that would be interesting to learn more about in future studies.

Additional comments

There are some minor errors in spelling a grammar that can be easily addressed. Given the scope and broadness of the subject matter, I felt that the study was reported adequately overall.

Note: Annotations were done in Preview for Apple OS X so formatting issues may make them show up different if opened in a different program.

---

## Round 0.2 · accepted · Accept

· Academic Editor

Accept

Thank you for the responsive revision. The paper is accepted with a minor edit of the last sentence, adding the subject "we" to the main sentence: "Moreover, although we found an association between physical activity and violence, WE suggest future studies that investigate the relationship between these variables [CAPS added for attention, not to be printed so]."